**Mie–Raman-Fluorescence lidar observations of aerosols during pollen season in the north of France**

Igor Veselovskii[1], Qiaoyun Hu[2], Philippe Goloub[2], Thierry Podvin[2], Marie Choël[3], Nicolas Visez[4], Mikhail Korenskiy[1]

[1]*Prokhorov General Physics Institute of the Russian Academy of Sciences, Moscow, Russia.*

[2]*Univ. Lille, CNRS, UMR 8518 - LOA - Laboratoire d'Optique Atmosphérique, Lille, 59000, France*

[3]Univ. Lille, CNRS, UMR 8516 - LASIR - Laboratoire de Spectrochimie Infrarouge et Raman, 59000, Lille, France

[4]*Univ. Lille, CNRS, UMR 8522 - PC2A - Physicochimie des Processus de Combustion et de l'Atmosphère, 59000, Lille, France*

**Correspondence**: Igor Veselovskii (iveselov@hotmail.com)

**Abstract**

Multiwavelength Mie–Raman–fluorescence lidar of Lille University with the capability to measure three aerosol backscattering, two extinction coefficients and three linear depolarization ratios together with the fluorescence backscattering at 466 nm was used to characterize aerosols during the pollen season in the north of France for the period March – June 2020. The results of observations demonstrate that the presence of pollen grains in aerosol mixture leads to an increase of the depolarization ratio. Moreover, the depolarization ratio exhibits a strong spectral dependence increasing with wavelength, which is expected for the mixture containing fine background aerosols with low depolarization and strongly depolarizing pollen grains. High depolarization ratio correlates with the enhancement of the fluorescence backscattering, corroborating the presence of pollen grains. Obtained results demonstrate that simultaneous measurements of particle depolarization and fluorescence allows to separate dust, smoke particles and aerosol mixtures containing the pollen grains.

**1. Introduction**

Pollen grains represent a significant fraction of primary biological particles emitted from the biosphere into the atmosphere in certain seasons and locations (Fröhlich-Nowoisky et al., 2016). There has been a growing interest in pollen study in recent years, because they can affect human





health by causing allergy-related diseases and contribute to the cloud formation by acting as giant cloud condensation nuclei (CCN) (Diehl et al., 2001; Pope, 2010; D'Amato et al., 2014; Steiner et al., 2015; Lake et al., 2017; Mack et al., 2020). To investigate the processes of pollen transport and dispersion, the information about vertical distribution of pollen grains is needed, and this information can be obtained from lidar measurements. Pollen grains are large irregularly shaped particles of complicated morphology (Frenguelli, 2003), causing strong depolarization of the backscattered laser radiation, which provides a basis for their identification. The first profiling of pollen with depolarization lidar was reported by Sassen (2008, 2011). His measurements over Alaska revealed that linear depolarization ratio of birch pollen plumes at 0.694 nm can exceed 30%. Further studies of pollen with elastic backscatter lidar at 532 nm were reported by Noh et al. (2013 a,b) and by Sicard et al. (2016). Their measurements confirmed high depolarization ratio of pollen grains (particle depolarization ratios as high as 43% were observed for aerosol mixture containing *Platanus* and *Pinus* pollen). Moreover, pollen grains backscattering demonstrated strong diurnal cycle, being highest near the noon. The use of multiwavelength observations increases capability of lidar technique for aerosol characterization. In recent studies of Bohlmann et al. (2019) and Shang et al., (2020) measurements performed with Polly[XT] lidar allowed to estimate mean values of the lidar ratios (about 45 sr and 55 sr at 355 and 532 nm respectively for birch pollen grains). The decrease of extinction and backscattering Angstrom exponents (EAE and BAE) during pollen episodes was also reported.

Atmospheric biological particles efficiently produce wideband fluorescence emission, when being exposed to UV radiation (Pohlker et al., 2012; Pan, 2015; Miyakawa et al., 2015), which offers an opportunity for monitoring them with fluorescence lidars. Nowadays, lidar spectrometers based on multianode photomultipliers allow a simultaneous detection of fluorescence backscattering in 32 spectral bins (Sugimoto et al., 2012; Reichardt et al., 2014, 2017; Saito et al., 2018). In particular, such lidar spectrometer was used in recent work of Saito et al. (2018) for remote measurement of the fluorescence spectrum of atmospheric pollen grains. The results demonstrate that, for 355 nm stimulating wavelength, the fluorescence spectra of different pollen grains have maxima in the 400–600 nm range and the intensity peak at around 460 nm.

To achieve the highest sensitivity of fluorescence detection, in many tasks it is preferable to use a single channel monitoring, where a part of the fluorescence spectrum is selected with a wideband interference filter (Immler et al, 2005; Rao et al., 2018; Li et al., 2019). In our recent





publication (Veselovskii et al., 2020) we reported the results obtained from a modified Mie-Raman
lidar (LILAS lidar system in Laboratoire d'optique Atmosphérique) with one additional
fluorescence channel at 466 nm. Such an approach has proved high sensitivity, allowing to detect
fluorescence signals from weak aerosol layers and to calculate the fluorescence backscattering
coefficient from the ratio of fluorescence and nitrogen Raman backscatters, thus making it
potentially attractive for pollen monitoring.
In the present research we combine capability of multiwavelength Mie–Raman lidar for
providing three backscattering, two aerosol extinction coefficients and linear depolarization ratio
at three wavelengths with single channel fluorescence measurements for characterization of
aerosol mixtures containing pollen grains. The measurements reported were performed during
March–June 2020 period at the Lille Atmospheric Observation Platform (https://www-loa.univ-
lille1.fr/observations/plateformes.html?p=apropos) hosted by Laboratoire d'Optique
Atmospherique, University of Lille, Hauts-de-France region.

**2. Instrumentation**
**2.1 Mie-Raman-Fluorescence lidar**
The measurements were performed using LILAS lidar system – a multiwavelength Mie-
Raman lidar, based on a tripled Nd:YAG laser with a 20 Hz repetition rate and pulse energy of 70
mJ at 355 nm. The backscattered light is collected by a 40 cm aperture Newtonian telescope. The
full geometrical overlap of the laser beam and the telescope FOV is achieved at approximately
1000 m and to obtain the information about particles at lower altitudes, part of the measurements
were performed at an angle of 30 degrees to the horizon. The system is designed for a simultaneous
detection of elastic and Raman backscattering, allowing the so called $3\beta+2\alpha+3\delta$ data configuration,
including three particle backscattering ($\beta_{355}$, $\beta_{532}$, $\beta_{1064}$), two extinction ($\alpha_{355}$, $\alpha_{532}$) coefficients
along with three particle depolarization ratios ($\delta_{355}$, $\delta_{532}$, $\delta_{1064}$). The particle depolarization ratio $\delta$,
determined as a ratio of cross- and co-polarized components of the particle backscattering
coefficient, was calculated and calibrated the same way as described in Freudenthaler et al. (2009).
The description of the system can be found in the recent publication of Hu et al., (2019).
To perform fluorescence lidar measurements, the water vapor Raman channel at 408 nm
was replaced by a fluorescence channel, whose spectrum is captured by a wideband filter centered
at 466 nm and of 44 nm width (Veselovskii et al., 2020). The fluorescence measurements were



performed during night time only. The aerosol extinction and backscattering coefficients at 355
and 532 nm were calculated from Mie-Raman observations (Ansmann et al., 1992), while $\beta_{1064}$
was derived by the Klett method (Klett, 1985). The fluorescence backscattering coefficient $\beta_F$ is
calculated from the ratio of fluorescence and nitrogen Raman backscattering, as described in
Veselovskii et al. (2020). This approach allows to evaluate the absolute values of $\beta_F$, if the relative
sensitivity of the channels is calibrated and the nitrogen Raman scattering cross section is known.
Corresponding uncertainty we estimate to be below 50%. Parameters of detectors were not
changed during the campaign, so uncertainty of relative variations of $\beta_F$ was significantly lower
and was determined by the statistical errors of fluorescence measurements.
To characterize the efficiency of the fluorescence in respect to elastic scattering, the
fluorescence capacity
$$G_F = \frac{\beta_F}{\beta_{532}}$$          (1)
is also used. This parameter depends on the relative humidity (RH), so information about RH is
important for data analysis. Radiosonde measurements are used to monitor water vapor, as the
water vapor channel is replaced by the fluorescence channel in current lidar configuration. The
closest available radiosonde data are from Herstmonceux (UK) and Beauvecchain (Belgium)
stations, located 160 km and 80 km away from the observation site respectively. These radiosonde
data are not collocated with the lidar measurements, so only qualitative analysis of humidification
effects was possible.

**2.2 Pollen in situ sampling**
Airborne pollen grains and spores were collected by a Hirst-type volumetric sampler
(VPPS 2000, Lanzoni s.r.l). The pollen sampler was located on the campus of the University of
Lille (France) on the rooftop of a 20-m-high building where the lidar instrument was operated.
Ambient air was sampled at 10 L.min$^{-1}$ flow rate, allowing the impaction of pollen and spores on
an adhesive strip mounted on a rotating clockwork-driven drum. The impaction surface moves at
2 mm.h$^{-1}$ behind the entrance slit, allowing a temporal resolution of 2 hours. The adhesive strip
was substituted every 7 days after a full rotation of the drum, which is splitted into 7 parts, each
corresponding to a day of monitoring. And then they are fixed on a microscope glass slide with
gelatin and fuchsine dye. Pollen taxa were identified by light microscopy on the basis of their




characteristic shape and size. Airborne pollen concentrations were expressed as a daily and dual
hourly number of pollen grains per cubic meter of air.
Fig.1 shows the most abundant pollen taxa for the period from March to June 2020 in Lille.
These include: *Betula* (54.8% of total pollen taxa over the period), *Fraxinus* (8.2%), *Quercus*
(5.8%), *Urticaceae* (4.6%), *Salix* (4.5%) and *Cupressaceae* (4.1%). The same figure shows also
the fluorescence backscattering $\beta_F$ measured by lidar. The results presented are obtained by
averaging all available data during the night and the maximal values in 500 – 1000 m range are
shown. The highest fluorescence is observed in the end of March, when ash (*Fraxinus*) is the main
pollinator. The period of intense birch (*Betula*) pollination (3-15 April 2020) correlates also with
high $\beta_F$. Strong fluorescence observed for 5-10 May and 28 May–2 Jun periods, can be due to grass
(*Poaceae*) pollen contribution. By the end of June, $\beta_F$ decreases and becomes comparable with
fluorescence backscattering of background aerosol. From Fig.1, we can conclude that there is no
direct correlation between in situ and fluorescence lidar measurements, thus pollen observed in the
boundary layer by the lidar are probably transported from other regions. However, comparing lidar
and in situ observations, we should keep in mind, that maximum of pollen emission occurs near
the noon, while lidar measurements were performed in the night.

**3. Discrimination of pollen from other types of aerosol**

*3.1. Specific features of pollen containing aerosol mixture*

In contrast to the observations performed over Alaska (Sassen, 2008, 2011) or Finland
(Bohlmann et al., 2011), where pollen concentration was high due to boreal forests surrounding,
the pollen loading in the north of France is significantly lower. Long-term lidar and sun photometer
observations performed at Lille University demonstrate that local aerosol is mainly of continental
type, with predominance of the fine mode particles and low depolarization ratio. The emission of
large pollen grains, should lead to strong spectral dependence of the depolarization ratio, because
the backscattering at 1064 is less sensitive to the fine background particles than at shorter
wavelengths, thus particle depolarization ratio at 1064 nm ($\delta_{1064}$) should be more sensitive to the
presence of pollen grains, compared to $\delta_{355}$ and $\delta_{532}$. The particle depolarization ratio $\delta$ of the
mixture, containing background aerosol (*b*) and pollen (*p*), with corresponding depolarization
ratios $\delta^b$ and $\delta^p$, can be calculated as:





$$\delta = \frac{\left(\dfrac{\delta^p}{1+\delta^p}\right)\beta^p + \left(\dfrac{\delta^b}{1+\delta^b}\right)\beta^b}{\dfrac{\beta^p}{1+\delta^p} + \dfrac{\beta^b}{1+\delta^b}} \qquad (2)$$
Here total backscattering $\beta = \beta^b + \beta^p$.
To estimate the dependence of depolarization of the aerosol mixture on the contribution of
pollen to the backscattering coefficient $\dfrac{\beta_{532}^p}{\beta_{532}}$ at 532 nm, a simplified simulation was performed.
Assuming that the depolarization ratios of pollen and background aerosol are spectrally
independent and that $\delta^p$=30% while $\delta^b$=3%, the mixture depolarization ratios $\delta_{355}$, $\delta_{532}$, $\delta_{1064}$ were
calculated as a function of $\dfrac{\beta_{532}^p}{\beta_{532}}$ using expression (2). The backscattering Angstrom exponents for
background aerosol are assumed to be $A_{355/532}^\beta = A_{532/1064}^\beta = 1.5$ , while for pollen
$A_{355/532}^\beta = A_{532/1064}^\beta = 0$. Results of computation are shown in Fig.2. For low $\dfrac{\beta_{532}^p}{\beta_{532}}$ the depolarization
ratio $\delta_{1064}$ significantly exceeds $\delta_{355}$ and $\delta_{532}$. Spectral properties of the real mixture can be more
complicated, due to possible spectral dependence of both $\delta^p$ and $\delta^b$. Information on laboratory
measured spectral dependence of depolarization ratios of pollen is rare. Cao et al. (2010) measured
the linear depolarization ratio of several types of pollen in a chamber at 355, 532 and 1064 nm
wavelengths. The results demonstrate a strong variation of spectral dependence for different taxa,
and for most of the samples $\delta_{1064}$ exceeded $\delta_{355}$. However, we should keep in mind that
measurements in the chamber were performed at low RH, and depolarization ratios at higher RH
may be different. Still, from Fig.2 we conclude that the increase of the particle depolarization ratio
with wavelength can be an indication of the presence of large, irregularly-shaped pollen grains.
We should recall that similar spectral dependence can be provided also by the dust particles in the
aerosol mixture. However, as it will be shown later, pollen possesses significantly higher
fluorescence capacity and this is how these particles can be discriminated from dust.
The presence of pollen should lead also to the decrease of the extinction and backscattering
Angstrom exponents. EAE depends mainly on particle size, while BAE is sensitive also to the
particle complex refractive index and shape, thus the measured profiles of EAE and BAE can
present significant difference (Veselovskii et al., 2015). In our study we analyze the EAE and BAE

179 for the wavelength pair 355/532 nm ( $A^{\alpha}_{355/532}$ and $A^{\beta}_{355/532}$ ) only, because the extinction and

180 backscattering coefficient involved are calculated from Mie–Raman observations.

181  When analyzing Mie-Raman-fluorescence lidar measurements of pollen containing aerosol

182 mixtures, the numerous factors should be taken into account. These factors include the

183 fluorescence of background aerosol and other non-pollen aerosols that have strong fluorescence

184 capacity, for example, smoke particles. Dust particles can contribute to the increase of

185 depolarization ratio and, finally, the hygroscopic growth can modify the particle parameters. All

186 these factors will be considered in following sections.

188  *3.2. Characteristics of background aerosol over observation site*.

189  Long-term observations in Lille University demonstrate that aerosol over the observation

190 site is mainly of continental type with predominance of the fine mode particles. Typical vertical

191 profiles of the background aerosol parameters, observed on 3 June 2020, are given in Fig.3,

192 showing aerosol elastic and fluorescence backscattering coefficients, lidar ratios, Angstrom

193 exponents and depolarization ratios at three wavelengths. The RH from Beauvecchain (Belgium)

194 radiosonde observations, was below 50% in the height range considered. Particle depolarization

195 ratios at all three wavelengths are below 7%, indicating that contribution of pollen to the total

196 backscattering is insignificant. This agrees with the low values of pollen concentration provided

197 by in situ measurements (Fig.1a). The lidar ratios at both wavelengths ($S_{355}$, $S_{532}$) are close, varying

198 in the 50-60 sr range, and the fluorescence capacity $G_F$ is below $0.35 \times 10^{-4}$. The EAE and BAE

199 ( $A^{\alpha}_{355/532}$ , $A^{\beta}_{355/532}$ ) are in the 1.5–2.0 range. The presence of pollen should lead to a deviation of the

200 particle intensive parameters, such as the fluorescence capacity, depolarization ratio, EAE and

201 BAE, from the typical values of background aerosol.

203  *3.3 Identification of the smoke particles*

204  During the observation period the smoke elevated layers transported over Atlantic were

205 frequently detected. Smoke particles, are characterized by high fluorescence cross section

206 (Reichardt et al., 2017; Veselovskii et al., 2020) and can interfere pollen fluorescence

207 measurements. The temporal evolution of the range corrected lidar signal, volume depolarization

208 ratio at 1064 nm and fluorescence backscattering for the smoke episode on the night 23 – 24 June

209 2020 are shown in Fig.4. During the night the smoke layer with low depolarization and high



fluorescence is observed at approximately 5000 m height. Back trajectories (not shown) indicate
that the layer is transported from Canada. Vertical profiles of the particle parameters for this
episode are shown in Fig.5. The lidar ratio is about 50 sr at 355 nm, while the lidar ratio at 532 nm
increases within the smoke layer from 60 sr to 80 sr. This increase of $S_{532}$ occurs simultaneously
with decrease of $A^{\alpha}_{355/532}$ from 1.5 to 0.75, indicating that the particle size inside the layer growths
with height. Higher values of $S_{532}$ in respect to $S_{355}$ are typical characteristics for the aged smoke
(e.g. Müller et al., 2005; Nicolae et al., 2013; Hu et al., 2019). The depolarization ratio decreases
with wavelength from $\delta_{355}$=10% to $\delta_{1064}$=1.5%. Strong spectral dependence of depolarization ratio
and, in particular, low values of $\delta_{1064}$, are the features, allowing to identify the smoke layers. The
extinction Angstrom exponent $A^{\alpha}_{355/532}$ in the center of layer is about 0.75, while $A^{\beta}_{355/532}$ is about
1.9 and shows no significant variation through the layer. High values of $A^{\beta}_{355/532}$ compared to
$A^{\alpha}_{355/532}$ is another feature that will be used for aged smoke discrimination. Smoke fluorescence
capacity is high, reaching up to $G_F$=5*10$^{-4}$ for the period of observations, and this is one more
feature, allowing to separate smoke from other types of aerosol.
*3.4 Identification of the dust particles*
Presence of dust particles and pollen in the fine background aerosol leads to some common
characteristics in the lidar data, such as the decrease Angstrom exponents and increase of
depolarization ratios. However, pollen and dust can be separated by the fluorescence capacity. The
vertical profiles of particle parameters during dust episode on 27 May are shown in Fig.6. The dust
containing layer extends from 2000 m to 7000 m and the particle depolarization ratios $\delta_{1064}$ and
$\delta_{532}$ in this layer are close to 20%. These values are lower than depolarization of pure dust. For
example, Freudenthaler et al. (2009) for pure dust provide the values of 27% and 31% at 1064 nm
and 532 nm wavelengths respectively, thus in our case transported dust particles may be mixed
with local aerosols. The particle depolarization at 355 nm is not shown in the figure, because the
scattering ratio in the dust layer was too low to compute $\delta_{355}$ reliably. The fluorescence capacity
of particles in the dust layer is about 0.1×10$^{-4}$ at 4000 m, which is factor 50 lower than $G_F$ of the
smoke in Fig.5. There is also a weak aerosol layer at 1600 m with $\beta_{532}$ about 0.035 Mm$^{-1}$sr$^{-1}$. The
fluorescence capacity in this layer is high ($G_F$≈2.0*10$^{-4}$), suggesting that this layer may contain
smoke or pollen particles.






### *3.5 Impact of particle hygroscopic growth*


The vertical variation of observed aerosol properties may be a result of particle water
uptake, which should be separated from the features related to pollen presence. Fig.7 shows the
profiles of the particle parameters for the episode on 15 June 2020, when the aerosol hygroscopic
growth could take place. In the height range 900–1500 m the fluorescence backscattering $\beta_F$ is
stable, while the elastic backscattering $\beta_{532}$ increases by a factor 3. Radiosonde measurements in
Herstmonceux (UK) in this height range demonstrate an increase of RH from about 75% to 85%,
while lidar measured extinction and backscattering Angstrom exponents decrease from 1.5 to 1.3,
corroborating the presence the particle hygroscopic growth. The depolarization ratio $\delta_{1064}$ at low
altitudes exceeds $\delta_{355}$ and $\delta_{532}$, which can be an indication of pollen presence. This is supported by
significant fluorescence capacity ($G_F=0.9\times10^{-4}$ at 750 m).
The number of fluorescent particles in the 900–1500 m range, does not present significant
changes ($\beta_F$ is stable), so observed vertical variations, i.e. the decrease of depolarization ratios at
all three wavelengths and the increase of lidar ratios $S_{355}$ and $S_{532}$ from 50 sr to 65 sr, are probably
the result of water uptake by the particles. Water uptake does not change the number of fluorescent
molecules, however the fluorescence capacity decreases in the process of the hygroscopic growth,
so $G_F$ can be a representative parameter of aerosol types only at the condition of low RH.

### 4. Results of lidar measurements in the presence of pollen


During March–June 2020, we had numerous measurement cases demonstrating the
features in the profiles of the particle parameters, that can be attributed to pollen. For representative
cases we have chosen observations with high depolarization ratio and high fluorescence
backscattering. The same time, we omitted the days with high relative humidity, to minimize the
impact of the hygroscopic growth effects. Below we consider several measurement cases
representing different scenarios, in particular, the episodes when pollen concentration decreases
with height (30-31 May, 1-2 June) and the episodes when pollen grains are well mixed inside the
boundary layer (27-28 March and 21 April).

### *4.1. 30-31 May and 1-2 June 2020 observations*




The results of lidar measurements during the campaign in many episodes can be interpreted
as decrease of pollen concentration with height. Vertical profiles of the main particle parameters
for two representative cases in the nights of 30-31 May and 1-2 June 2020, are shown in Fig.8.
The HYSPLIT back trajectory analysis (Stein et al., 2015) demonstrates that in 1000--2000 m
height range the air masses were transported from the Northern Europe. At the ground level, the
grass could be the main pollen contributor for this period, as shown in Fig 1. On 31 May (at 00:00
UTC) the RH measured by the radiosonde in Herstmonceux (UK) was about 40% at 500 m and it
increased up to 70% at 2000 m. On 2 June the RH increased from approximately 40% to 60% in
the same height range. For both nights the fluorescence backscattering decreases with height,
indicating the decrease of the concentration of fluorescent particles (presumably pollen). This
decrease of $\beta_F$ correlates with decrement of the depolarization ratio at all three wavelengths.
Particle depolarization $\delta_{1064}$ is the highest (about 15% at 750 m), while $\delta_{355}$ and $\delta_{532}$ are
significantly lower. Such spectral dependence of depolarization ratio agrees with model
calculation in Fig.2. The lidar ratios are available above 1250 m and for both cases, $S_{355}$ and $S_{532}$
increase with height. It indicates that the lidar ratios of pollen in the two considered cases can be
quite low: below 40 sr at 355 nm and below 30 sr at 532 nm, considering that pollen concentration
decreases with height, which is inferred from the features of depolarization ratio and fluorescence
backscattering
The EAE for both nights is about 2.0 and does not show significant changes with height.
The BAE is lower (about 1.5 at 1000 m) and for both nights it shows some increase in 1250–2250
m range. The BAE, in contrast to EAE, depends strongly on the particle refractive index and shape,
thus it may demonstrate higher sensitivity to the changes in aerosol mixture composition. Recall
that backscattering and extinction Angstrom exponents are related as:
$$A^{\beta}_{355/532} = A^{\alpha}_{355/532} + \frac{\ln(S_{532}/S_{355})}{\ln(355/532)} \qquad (3)$$
Thus for $S_{355} > S_{532}$, which has been observed during pollen episodes, the $A^{\beta}_{355/532}$ is lower than
$A^{\alpha}_{355/532}$. This is in contrast with smoke episodes, where $S_{355} > S_{532}$ and $A^{\beta}_{355/532} > A^{\alpha}_{355/532}$ (Fig.5).
If the depolarization ratios of pollen $\delta^p$ and background aerosol $\delta^b$ are known, the pollen
backscattering coefficient $\beta^p$ can be calculated. Such approach is widely used for the separation
of contributions of dust and smoke particles (Sugimoto and Lee, 2006; Tesche et al., 2009) and





the same technique was applied to separate pollen and background aerosol (Noh et al. 2013a;
Sicard et al., 2016; Shang et al., 2020). For height independent depolarization ratios of pollen and
background aerosol the pollen backscattering coefficient can be calculated as suggested by Tesche
et al. (2009):
$$\beta^p = \beta \frac{(\delta - \delta^b)}{(\delta^p - \delta^b)} \frac{(1+\delta^p)}{(1+\delta)} \quad , \tag{4}$$

Here $\beta$ and $\delta$ are backscattering coefficient and particle depolarization ratio of the mixture. The
profiles of $\beta^p_{532}$ and the relative contribution $\frac{\beta^p_{532}}{\beta_{532}}$ are shown in Fig.8(b,e). Computations were
performed in assumption of height independent $\delta^p_{532} = 30\%$. For background aerosol, the values
$\delta^b_{532} = 3\%$ for 30-31 May and $\delta^b_{532} = 5\%$ for 1-2 June were used. On $30 - 31$ May contribution of
pollen $\frac{\beta^p_{532}}{\beta_{532}}$ at 750 m is estimated as 30%. From Fig.2 the expected ratio $\delta_{1064}/\delta_{355}$ is about 2.3. It
agrees with observed ratio, which is about 2.4.

The profiles of $\beta_F$ and $\beta^p_{532}$ in Fig.8(b, e) behave similarly, decreasing with height. Above

2000 m the decrease of $\beta_F$ slows down due to the fluorescence of background aerosol. The profiles
of the fluorescence capacity $G_F$ and relative contribution $\frac{\beta^p_{532}}{\beta_{532}}$ also demonstrate a good correlation.
Thus both depolarization and fluorescence techniques lead to the same conclusion: pollen
concentration in the boundary layer for the considered episodes decreases with height.

***4.2. 27 – 28 March and 21 April 2020 observations.***

According to the in-situ pollen sampling at rooftop level, the maximal pollen content was

detected during birch pollination period on 4–20 April. However, the maximal fluorescence
backscattering of lidar data was observed in the end of March, when sampling shows an increase
of ash (*fraxinus*) pollen emission. The temporal evolution of range corrected lidar signal, volume
depolarization ratio at 1064 nm and fluorescence backscattering on 27-28 March night is shown
in Fig.9. The main part of the aerosol is localized below 2000 m. The back trajectory analysis
demonstrates the air masses in this episode were transported from the East Europe. In contrast with
Fig.8, where fluorescence decreases with height, on 27 -28 March the fluorescent particles are



rather well mixed inside the PBL (planetary boundary layer). The fluorescence backscattering is
high, exceeding $2.5 \times 10^{-4}$ $Mm^{-1}sr^{-1}$ and the volume depolarization at 1064 nm is above 15%. The
vertical profiles of the particle parameters are shown in Fig.10. Radiosonde measurements (at both
Beauvecchain and Herstmonceux sites) show that RH gradually increased with height from
approximately 40% to 70% in 500–1750 m range.
Both the fluorescence backscattering and depolarization ratios do not demonstrate strong
variations inside the 600–1500 m range. The maximum of fluorescence capacity exceeds $1.2 \times 10^{-4}$,
which is significantly higher than $G_F$ for background aerosol in Fig.3. The profiles of $G_F$ and
$\dfrac{\beta_{532}^{p}}{\beta_{532}}$ behave reasonably similar, the slight decrease of $\dfrac{\beta_{532}^{p}}{\beta_{532}}$ with height in respect to $G_F$ can be
due to dependence of depolarization ratio of pollen on RH.
Agreements between results obtained from depolarization and fluorescence techniques in
Fig.8,10, corroborates the suggestion that the observed fluorescence is mainly due to the presence
of pollen. However, in some episodes the particles with high fluorescence cross section, other than
pollen, could interfere. In particular, such interference occurred in 20-23 April 2020 period. Fig.11
shows the vertical profiles of particle parameters measured on 21 April. The depolarization ratio
$\delta_{1064}$=22% at 750 m was one of the highest during campaign. The RH is low, increasing from 30%
to 45% in 800–1500 m range, according to Herstmonceux radio sounding. The back trajectory
analysis demonstrates, below 1500 m the air masses are transported from Spain, while at 2000 m
the transportation is from the Northern Europe.
Fluorescence backscattering is stable in 500–1500 m range and the fluorescence capacity
at 1000 m is about $1.5 \times 10^{-4}$, which is a typical value for the pollen. However, above 1250 m $G_F$
starts to rise, reaching the value of $2.5 \times 10^{-4}$ at 1750 m, meanwhile depolarization ratio decreases.
Such high $G_F$ is more typical for the smoke particles. The lidar ratios above 1250 m, as well as
BAE, also increase. Such vertical variation of the intensive particle parameters was observed
during 20-23 April period and it may indicate the presence of the biomass burning aerosol near the
boundary layer top.

*4.3. Separation of pollen and smoke layers*
During the campaign we observed narrow layers with strong fluorescence. Two examples
of such observations, in the nights 13-14 April and 16-27 May 2020, are shown in Fig.12. The
white arrows on this figure point to the fluorescent layers. On 13 April, a weak aerosol layer
($\beta_{532} \approx 0.6$ Mm$^{-1}$ for 23:00 – 00:00 UTC) is observed at the top of the PBL. This layer demonstrates
volume depolarization ratios exceeding 10% and high fluorescence backscattering. On 26-27 May
a weak layer with high fluorescence backscattering occurs between 3 km and 4 km. However, in
contrast with the first case, it has low depolarization ratio, so the layers may have different nature.
Fig.13 shows the vertical profiles of the particle parameters for these two cases. On 13-14 April
the fluorescence backscattering below 1000 m is stable, while $\beta_{532}$ rises, which can be the result
of the particle water uptake. Above 1000 m, the depolarization ratio $\delta_{1064}$ increases up to 8%.
Results in Fig.13a are averaged over 21:15–00:40 UTC temporal interval, but peak values of $\delta_{1064}$
between 23:00 and 00:00 exceeded 12%. Fluorescence backscattering increases simultaneously
with the depolarization. The aerosol backscattering coefficient of fluorescent layer is too low for
a reliable calculation of $\delta_{355}$ and $\delta_{532}$, so only the profile of $\delta_{1064}$ is provided.
On 26-27 May the backscattering coefficient of the fluorescent layer at 3400 m is lower
than in Fig.13a ($\beta_{532} \approx 0.14$ Mm$^{-1}$sr$^{-1}$), so the depolarization ratio $\delta_{1064}$ can be calculated only in the
center of the layer and it is about 2%, which is significantly lower than that on 13-14 April.
However, the fluorescence capacity on 26-27 May is up to $3.5 \times 10^{-4}$, which is typical for smoke.
Thus, we can conclude that the fluorescent layer on 26-27 May contains the smoke particles, due
to high $G_F$ and low $\delta_{1064}$. On 13-14 April, the fluorescence capacity is significantly lower (about
$0.9 \times 10^{-4}$) and depolarization ratio $\delta_{1064}$ exceeds 10%, which is more typical for pollen. Due to low
backscattering coefficients of the fluorescent layers in Fig.12, we are not able to provide a
complete set of intensive parameters, such as Angstrom exponents and particle depolarization
ratios at three wavelengths. However, based on the obtained fluorescence capacities and $\delta_{1064}$
values, we conclude that the fluorescent layers probably contain pollen grain in Fig.12a, and smoke
particles in Fig.12b.

***4.4 Aerosol classification based on polarization and fluorescence measurements.***
Table 1 summarizes the results in the campaign, showing the aerosol parameters, such as
particle depolarization and lidar ratios, extinction Angstrom exponent, fluorescence backscattering
and capacity for several days in March–June 2020 observations period, when the contribution of
pollen to the total particle backscattering was significant. All available night observations were





averaged and results are given for heights with the highest particle depolarization. Lidar ratios
varied approximately in 40--70 sr range, wherein normally $S_{355}$ is greater than $S_{532}$. It must be
emphasized that pollen lidar ratios may differ for different taxa and that the observed lidar ratios
are not attributed to pure pollen, but to the aerosol–pollen mixture, so the values provided are
influenced by the properties of background aerosol. Moreover, the shape of pollen grains depends
on RH, (Heidemarie et al., 2018), which may also lead to the variation of pollen lidar ratios. In
most of the cases, the depolarization ratio presents strong spectral dependence and increases with
wavelength. This spectral dependence is probably the result of mixing of strongly depolarizing
pollen grains with fine background aerosol. The maximal value of observed fluorescence capacity
of pollen-aerosol mixture is $1.6 \times 10^{-4}$, which is significantly higher than that of background aerosol,
but lower than fluorescence capacity of smoke.
The simultaneous observations of depolarization ratio and fluorescence capacity for
different types of aerosol are summarized by Fig.14. On this plot, particle depolarization $\delta_{532}$ is
plotted versus $G_F$. The diagram allows to separate four types of the particles: (i) dust particles –
high $\delta_{532}$ and low $G_F$; (ii) pollen – high $\delta_{532}$ and high $G_F$; (iii) smoke – low $\delta_{532}$ and low $G_F$; (iiii)
background aerosol (continental type) - low $\delta_{532}$ and low $G_F$. Points corresponding to the pollen
mixture provide extended pattern, because parameters depend on the concentration of pollen in the
aerosol mixture. The dust measurements are also scattered, because dust over the instrumentation
site is long transported and mixed with local aerosol. Minimum $G_F$ for dust is about $0.1 \times 10^{-4}$ while
for smoke maximal $G_F$ is about factor 50 higher. The fluorescence capacity depends on the relative
humidity, so strong scattering of measurement points can be partly also due to RH variations.
Maximal values of $G_F$ for pollen mixture were about $1.5 \times 10^{-4}$, and the corresponding
depolarization ratios $\delta_{532}$ are about 18%. Thus, assuming that depolarization ratio of pure pollen is
30%, we can expect $G_F$ for pure pollen to be about $2.5 \times 10^{-4}$, which is comparable with values for
smoke.

**Conclusion**
We analyzed the measurements from a multiwavelength Mie-Raman-fluorescence lidar
during March–June 2020 in the north of France, to reveal the features that can be attributed to
pollen grains. Contrary to previous studies, where pollen was identified by the enhanced
depolarization ratio at a single wavelength, our lidar system allowed to measure depolarization



ratios at three wavelengths, simultaneously with the fluorescence backscattering at 466 nm. In
numerous episodes during the campaign, high values of the particle depolarization ratio at 1064
nm, exceeding 15%, were observed. Moreover, depolarization ratio had strong spectral
dependence, being the highest at 1064 nm and lowest at 355 nm, which is expectable for big
particles of irregular shape mixed with fine, low depolarizing background aerosol. The increase of
particle depolarization correlated with enhancement of the fluorescence backscattering
corroborating that in these episodes we observed aerosol mixtures containing pollen.
The lidar ratios of aerosol–pollen mixtures observed during campaign varied in a wide
range. At low altitudes, where particle presented strong depolarization and fluorescence, in many
cases we observed lidar ratios below 40 sr at both wavelengths. However, we had also cases when
the lidar ratios at both wavelengths were in 50–60 sr range. Thus, at the moment we are not capable
to specify lidar ratios for pure pollen and additional measurement campaigns in the locations with
high pollen content are strongly desirable.
Obtained results demonstrate, that simultaneous measurements of particle depolarization
and fluorescence allows to separate dust, smoke particles and pollen grains. Moreover, the
fluorescence measurements provide additional information that can be used in aerosol
classification schemes. However, further studies are needed to make this technique applicable for
the quantitative pollen characterization. In the data analysis it is important to account for the
process of water uptake by the particles, because hygroscopic growth increases backscattering of
background aerosol and influences the pollen grain shape. In the presented lidar configuration, the
water vapor channel was absent and radiosonde RH data were not collocated with lidar, which
prevented us from performing a quantitative analysis of the hygroscopic effects. Since December
2020, we recovered the water vapor channel in upgraded configuration of the lidar. Moreover, we
added one more fluorescence channel centered at 549 nm, which will be used in the next pollen
campaign in 2021. This additional channel should improve discrimination of the pollen from other
aerosols. In coming campaign we will try to correlate our results with pollen concentration at
different location in Europe by using the transport model, e.g. SILAM (System for Integrated
modeLling of Atmospheric coMposition) (Sofiev et al, 2013, 2015). The use of this model should
help in identification of pollen type in our observations.

**Acknowledgement**



We acknowledge funding from the CaPPA project funded by the ANR through the PIA under
contract ANR-11-LABX-0005-01, the "Hauts de France" Regional Council and the European
Regional Development Fund (ERDF). The "Réseau National de Surveillance Aérobiologique"
(RNSA) and the "Association pour la Prévention de la Pollution Atmosphérique" (APPA) are
gratefully acknowledged for providing Hirst-collected pollen grains identification and for
assistance with the pollen data handling.






Table 1. Lidar measured aerosol parameters, such as particle depolarization ratios ($\delta_{355}$, $\delta_{532}$, $\delta_{1064}$),
lidar ratios ($S_{355}$, $S_{532}$), extinction Angstrom exponent ($A^{\alpha}_{355/532}$), fluorescence backscattering
coefficient ($\beta_F$) and fluorescence capacity ($G_F$) for several days during March – June 2020 period,
when contribution of pollen to the total particle backscattering could be significant.

| Day | Height, m | $\delta_{355}$, % | $\delta_{532}$, % | $\delta_{1064}$, % | $S_{355}$, sr | $S_{532}$, sr | $A^{\alpha}_{355/532}$ | $\beta_F*10^{-4}$, Mm$^{-1}$sr$^{-1}$ | $G_{F*}10^{-4}$ |
|---|---|---|---|---|---|---|---|---|---|
| 27 Mar | 1150 | 9 | 12 | 13 | 50 | 42 | 1.5 | 2.5 | 1.2 |
| 7 Apr | 1150 | 13 | 13 | 13 | 60 | 60 | 1.25 | 1.0 | 1.3 |
| 8 Apr | 1000 | 11 | 10 | 9 | 50 | 60 | 1.0 | 1.9 | 1.0 |
| 15 Apr | 750 | 15 | 15 | 17 | 40 | 40 | 0.7 | 0.4 | 0.9 |
| 16 Apr | 1250 | 15 | 15 | 15 | - | - | - | 0.6 | 1.6 |
| 19 Apr | 650 | 8 | 10 | 14 | 58 | 48 | 1.35 | 1.2 | 0.9 |
| 20 Apr | 1000 | 18 | 18 | 22 | 55 | 45 | 1.2 | 0.75 | 1.3 |
| 21 Apr | 750 | 15 | 17 | 22 | 66 | 47 | 1.25 | 1.1 | 1.4 |
| 22 Apr | 1000 | 18 | 18 | 22 | 70 | 55 | 1.2 | 0.9 | 1.5 |
| 23 Apr | 1000 | 6 | 14 | 11 | 53 | 65 | 1.25 | 1.5 | 1.05 |
| 30 May | 750 | 7 | 10 | 16 | - | - | - | 0.8 | 1.2 |
| 1 June | 750 | 7 | 10 | 16 | - | - | - | 1.25 | 1.5 |




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

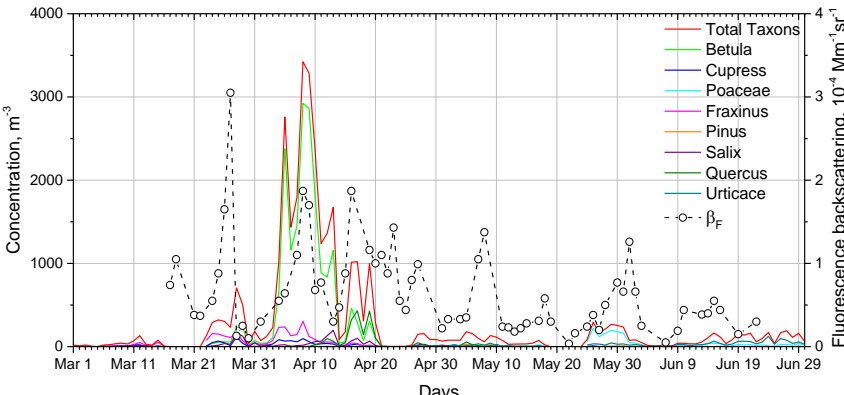


Fig.1. Daily concentration of most abundant pollen taxa, for the period March–June 2020 in Lille
from in situ measurements on the rooftop. Open symbols show fluorescence backscattering $\beta_F$
measured by lidar. Lidar measurements are averaged over night and maximal value in 500–1000
m range is shown.







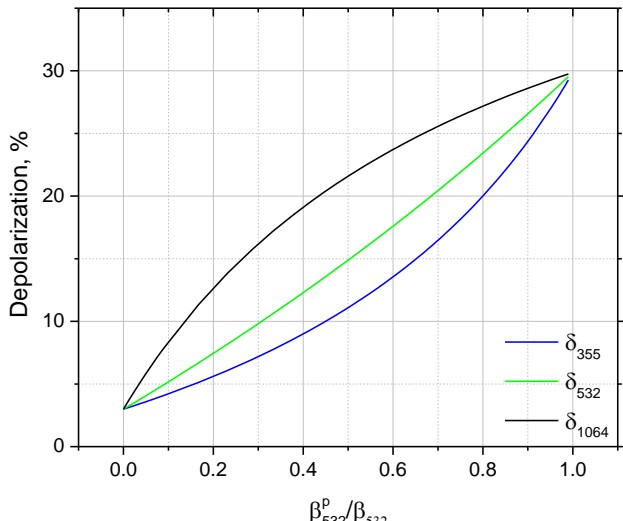


Fig.2. Depolarization ratios at 355, 532 and 1064 nm as a function of pollen contribution to the
total backscattering coefficient $\dfrac{\beta^p_{532}}{\beta_{532}}$. Depolarization ratios of pollen ($\delta^p$) and background aerosol
($\delta^b$) are assumed to be spectrally independent, with values of $\delta^p=0.3$ and $\delta^b=0.03$. The
backscattering coefficient of pollen is spectrally independent, while for background aerosol the
backscattering Angstrom exponents $A^{\beta}_{355/532} = A^{\beta}_{532/1064} = 1.5$ were used.



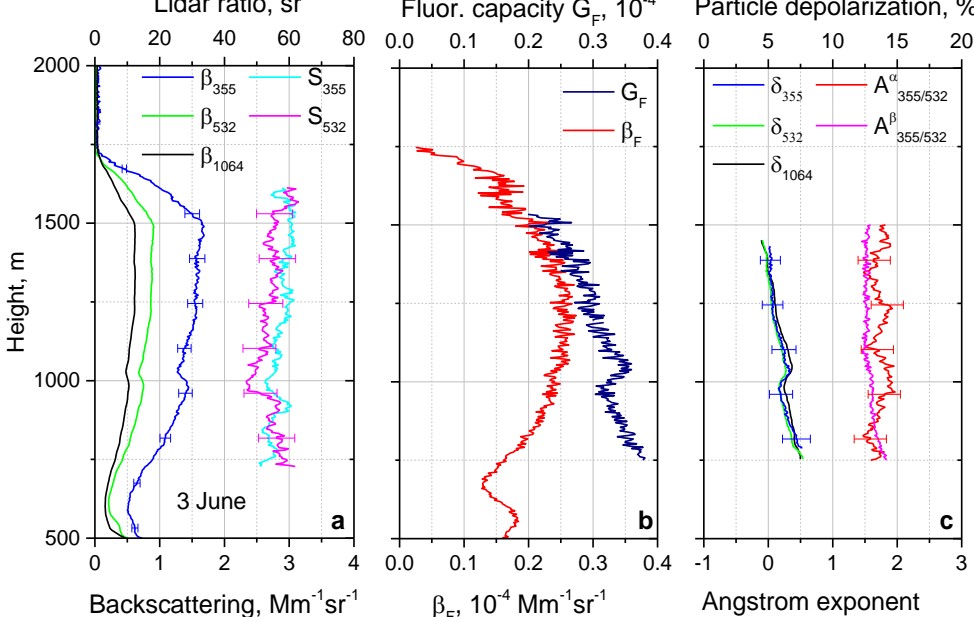


Fig.3. Measurements in the condition of background aerosol predominance. Vertical profiles of (a)
backscattering coefficients $\beta_{355}$, $\beta_{532}$, $\beta_{1064}$ and lidar ratios $S_{355}$, $S_{532}$; (b) fluorescence
backscattering $\beta_F$ and fluorescence capacity $G_F = \beta_F/\beta_{532}$; (c) particle linear depolarization ratios
$\delta_{355}$, $\delta_{532}$, $\delta_{1064}$ together with extinction and backscattering Angstrom exponents $A^{\alpha}_{355/532}$, $A^{\beta}_{355/532}$
on 3 June 2020 for 20:30 – 23:00 UTC. Measurements were performed at 30 degree to the horizon.







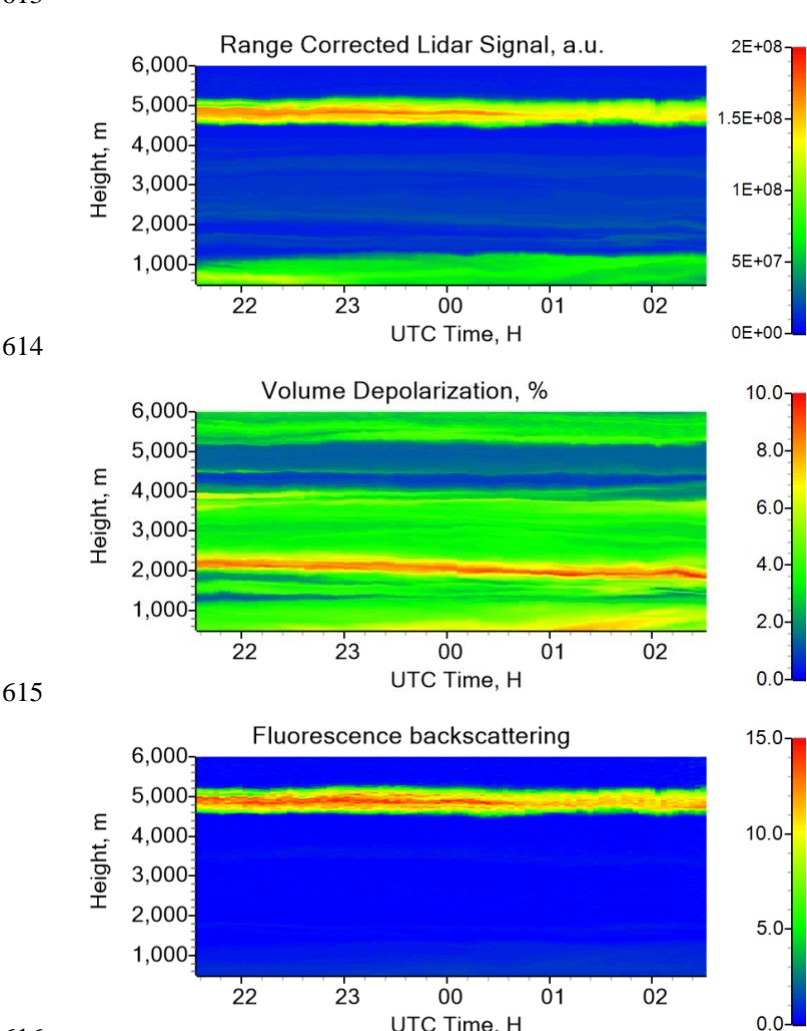




Fig.4. Range corrected lidar signal at 1064 nm, volume depolarization ratio at 1064 nm and fluorescence backscattering coefficient (in $10^{-4}$ Mm$^{-1}$sr$^{-1}$) on 23-24 June 2020.




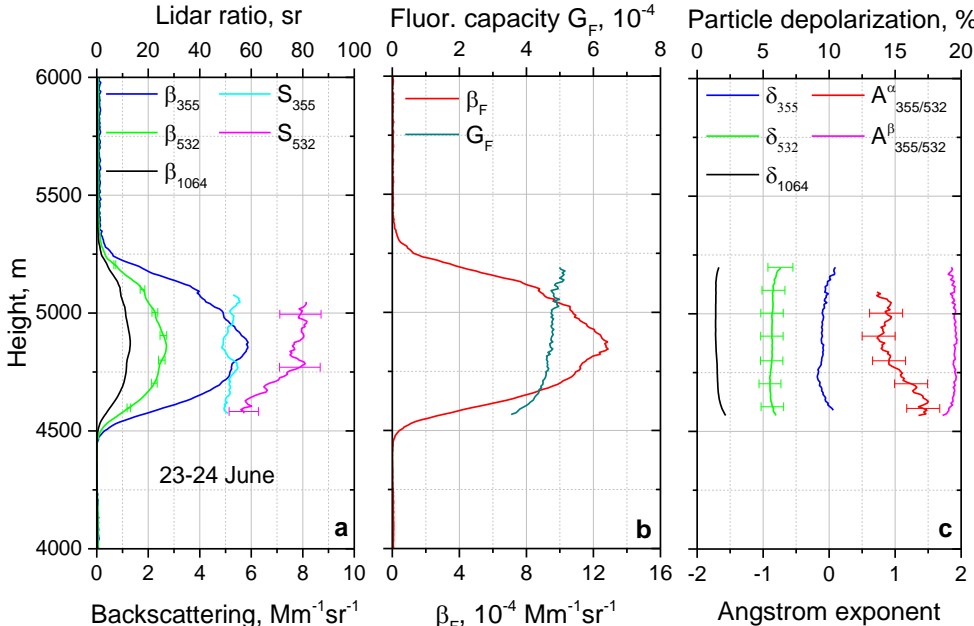


Fig.5. Vertical profiles of (a) backscattering coefficients $\beta_{355}$, $\beta_{532}$, $\beta_{1064}$, lidar ratios $S_{355}$, $S_{532}$; (b)
fluorescence backscattering coefficient $\beta_F$, fluorescence capacity $G_F$; and (c) particle
depolarization ratios $\delta_{355}$, $\delta_{532}$, $\delta_{1064}$ together with the extinction and backscattering Angstrom
exponents $A^{\alpha}_{355/532}$, $A^{\beta}_{355/532}$ on the night 23-24 June 2020 for 21:30 – 02:30 UTC.







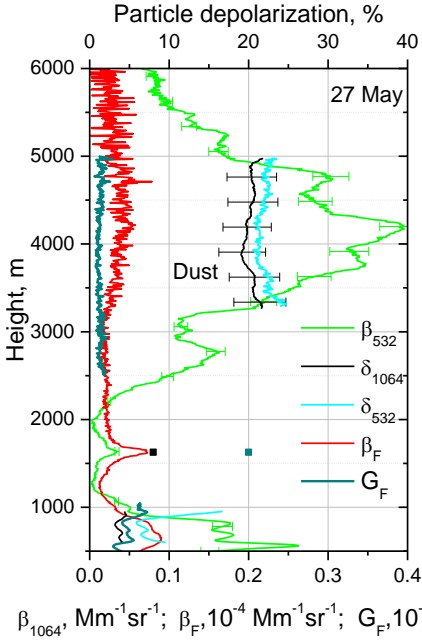


Fig.6. Lidar measurements during dust episode. Vertical profiles of particle $\beta_{1064}$ and fluorescence
$\beta_F$ backscattering coefficients, fluorescence capacity $G_F$ and particle depolarization ratios $\delta_{1064}$,
$\delta_{532}$ on 27 May 2020 for 21:00–23:00 UTC.








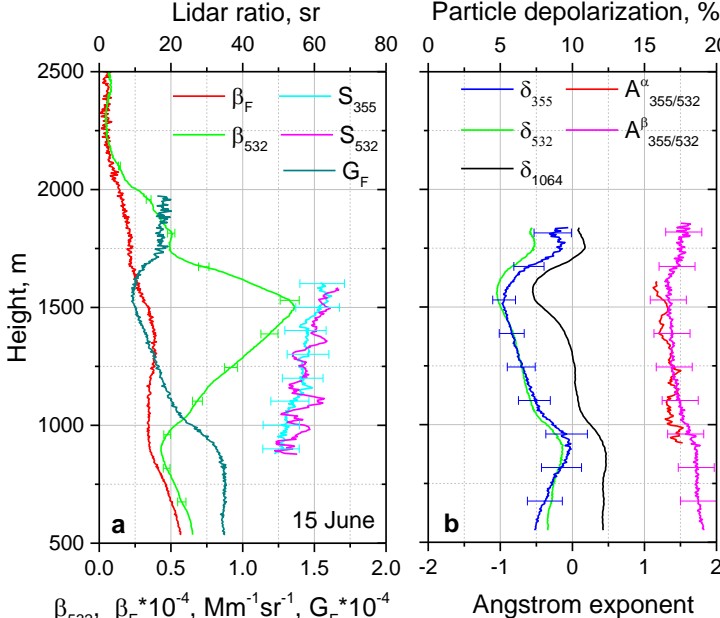


Fig.7. Lidar measurements in the condition of the aerosol hygroscopic growth in 900-1500 m
height range. Vertical profiles of (a) particle $\beta_{532}$ and fluorescence $\beta_F$ backscattering coefficients,
fluorescence capacity $G_F$, lidar ratios $S_{355}$, $S_{532}$ and (b) particle depolarization ratios $\delta_{355}$, $\delta_{532}$, $\delta_{1064}$
together with extinction $A^{\alpha}_{355/532}$ and backscattering $A^{\beta}_{355/532}$ Angstrom exponents measured on 15
June 2020 for 22:00 – 24:00 UTC.




Fig.8. Vertical profiles of (a, d) particle backscattering coefficients $\beta_{355}$, $\beta_{532}$, $\beta_{1064}$ and lidar ratios
$S_{355}$, $S_{532}$, (b, e) fluorescence backscattering coefficient $\beta_F$, fluorescence capacity $G_F$, pollen





backscattering coefficient $\beta^p_{532}$ and its contribution to the total backscattering $\dfrac{\beta^p_{532}}{\beta_{532}}$ ; (c, f) particle
depolarization ratios $\delta_{355}$, $\delta_{532}$, $\delta_{1064}$ together with extinction $A^{\alpha}_{355/532}$ and backscattering $A^{\beta}_{355/532}$
Angstrom exponents on (a-c) 30 - 31 May 2020 for 21:00 – 02:00 UTC and on (d-f) 1-2 June 2020
for 21:00 – 02:30 UTC. Profiles of $\beta^p_{532}$ and $\dfrac{\beta^p_{532}}{\beta_{532}}$ were computed in assumption of $\delta^p_{532} = 30\%$ .
The depolarization ratios of the background aerosol $\delta^b_{532}$ is measured/assumed to be 3% on 30 May
and 5% on 1 May.

















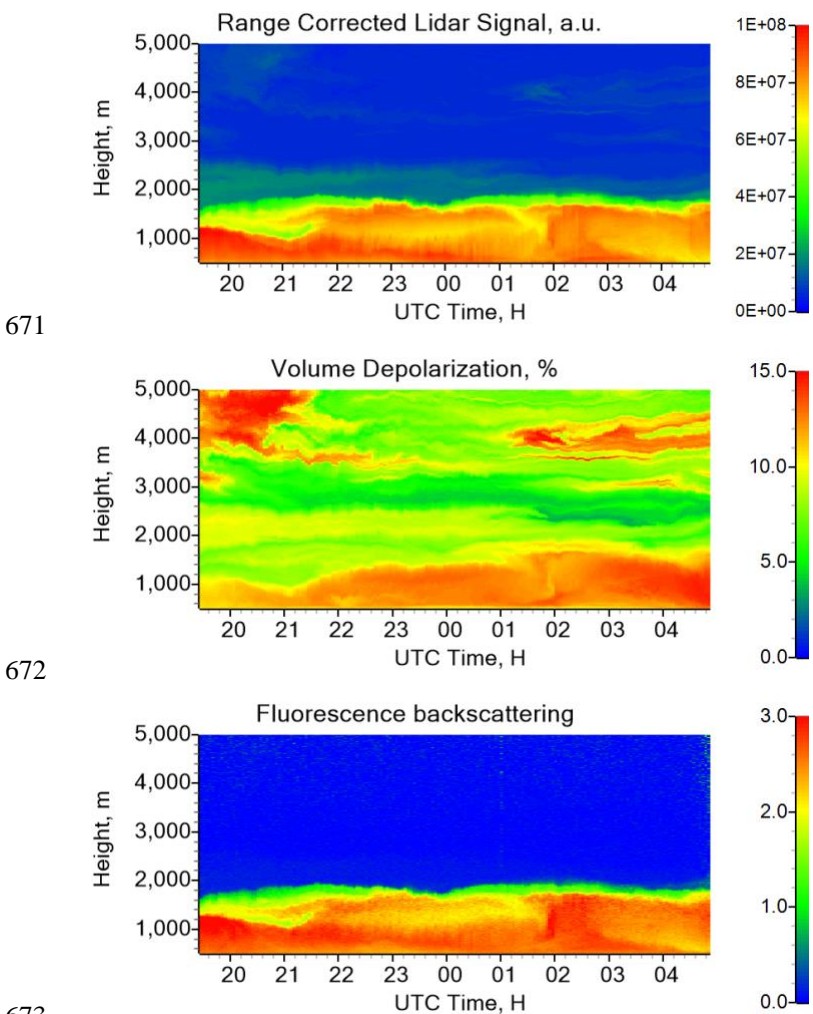




Fig.9. Range corrected lidar signal at 1064 nm (upper panel), volume depolarization ratio at 1064
nm (middle panel) and fluorescence backscattering coefficient (in $10^{-4}$ $Mm^{-1}sr^{-1}$, lower panel)
measured on 27-28 March 2020.




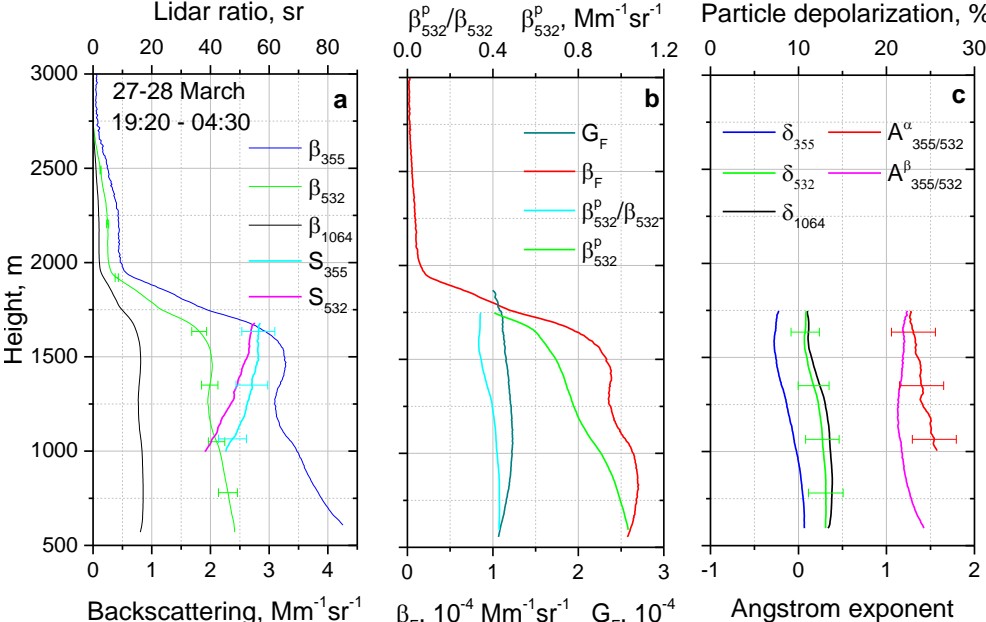


Fig.10. Vertical profiles of (a) particle backscattering coefficients $\beta_{355}$, $\beta_{532}$, $\beta_{1064}$ and lidar ratios
$S_{355}$, $S_{532}$; (b) fluorescence backscattering coefficient $\beta_F$, fluorescence capacity $G_F$, pollen
backscattering coefficient $\beta_{532}^{p}$ and its contribution to the total backscattering $\dfrac{\beta_{532}^{p}}{\beta_{532}}$; (c) particle
depolarization ratios $\delta_{355}$, $\delta_{532}$, $\delta_{1064}$ together with extinction $A_{355/532}^{\alpha}$ and backscattering $A_{355/532}^{\beta}$
Angstrom exponents measured on 27 - 28 March 2020 for19:20 – 04:30 UTC.






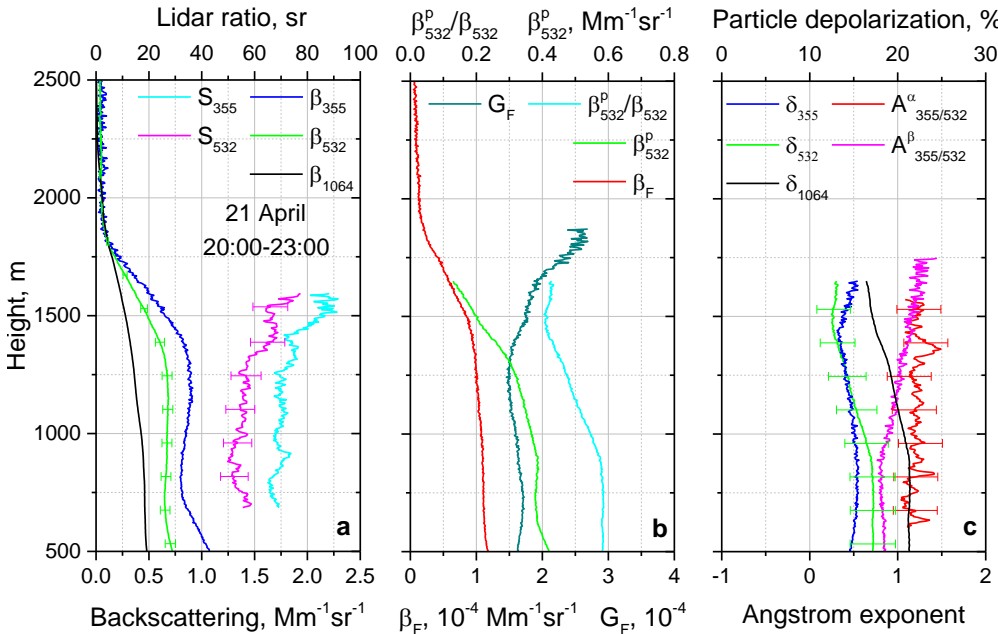


Fig.11. The same particle parameters as in Fig.10 for 21 April 2020, 20:00-23:00 UTC.
Measurements were performed at 30 deg to horizon.

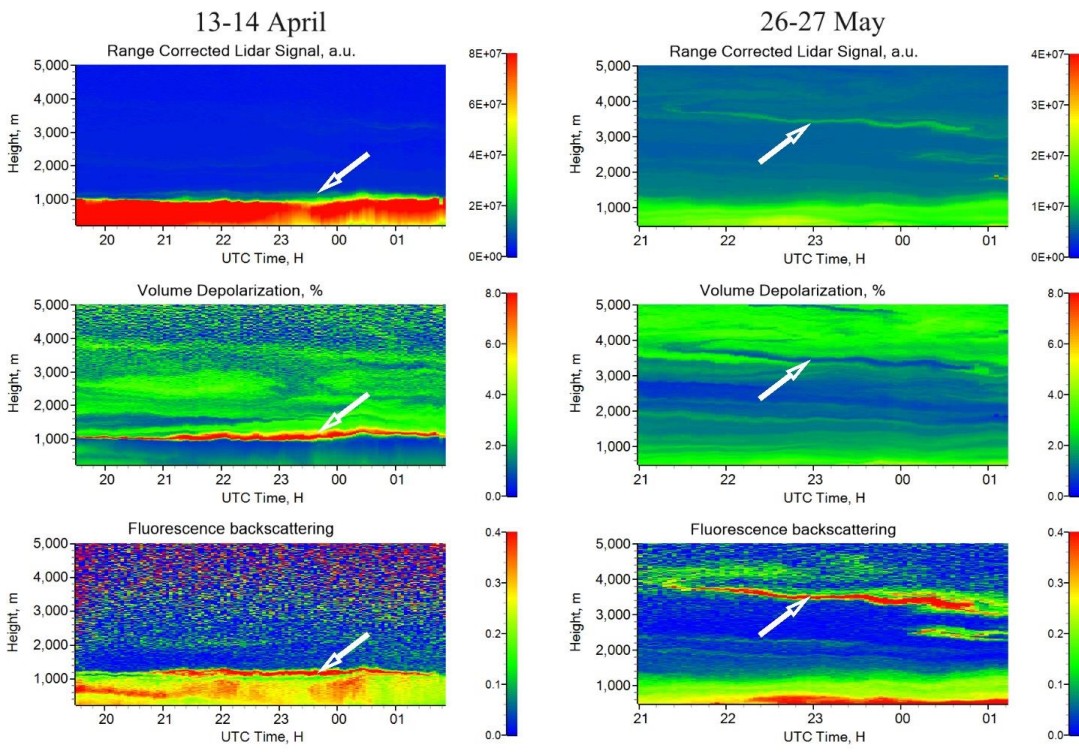


Fig.12. Range corrected lidar signal at 1064 nm, volume depolarization ratio $\delta^v_{1064}$ and
fluorescence backscattering coefficient (in $10^{-4}$ $Mm^{-1}sr^{-1}$) measured on 13-14 April (left column)
and 26-27 May 2020 (right column). Arrows point to the fluorescent layers.





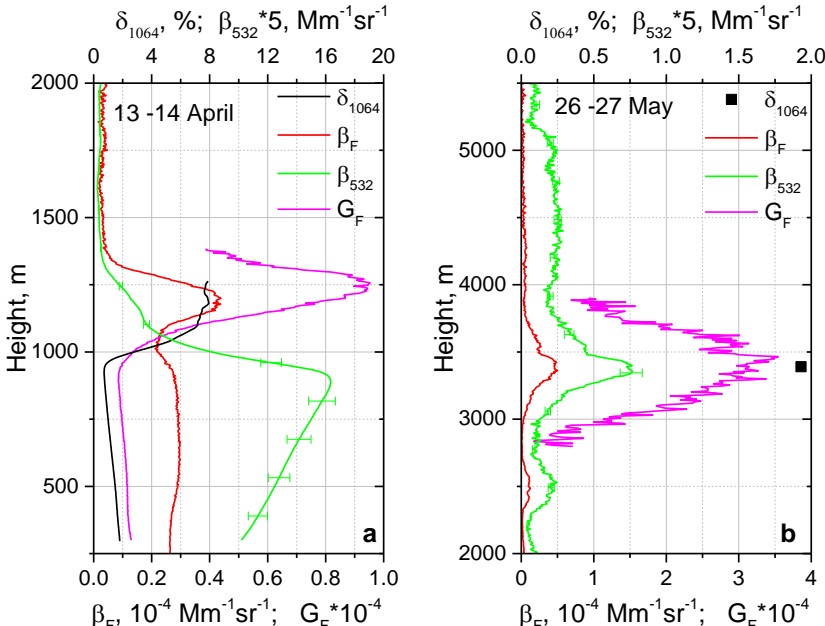


Fig.13. Vertical profiles of elastic $\beta_{532}$ and fluorescence $\beta_F$ backscattering coefficients,
fluorescence capacity $G_F$ and particle depolarization ratio $\delta_{1064}$ measured on (a) 13-14 April for
21:00 – 01:00 UTC and (b) 26-27 May 2020 for 23:30 – 00:40 UTC. Values of $\beta_{532}$ are multiplied
by factor 5.








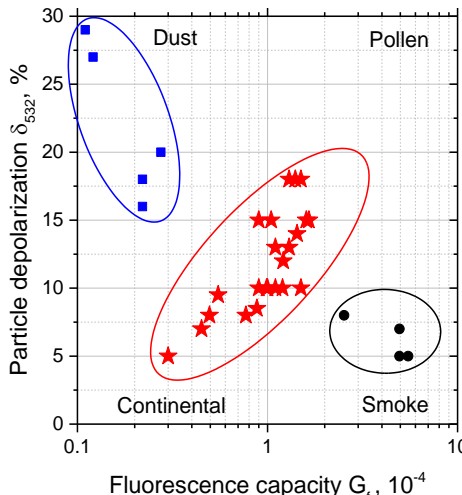


Fig.14. Particle depolarization $\delta_{532}$ versus fluorescence capacity $G_F$. This diagram allows to
identify dust (blue), smoke particles (black) and aerosol mixtures containing pollen (red).
