# Peer review of "Mie-Raman-Fluorescence lidar observations of aerosols during pollen season in the north of"

_Atmospheric Measurement Techniques, 2021_

## Author Comment (AC1)

**Response to Referee #1**

"This paper conducted research on pollen using the latest Lidar technology. Pollen is one of the major aerosols and is a particle that causes many health problems and is not easily measured. In particular, it is difficult to determine the distribution in the atmosphere. For this reason, observation using lidar is being performed, but studies using depolarization ratio, which can observe non-spherical features of pollen, have been mainly conducted. However, this paper confirms more information by measuring fluorescence at 466 nm with depolarization ratio. The thesis is judged to be well-written and there are no special modifications."

We are grateful to Ref#1 for positive assessment of our work

Technical comments;
1. In the figures, each parameter is separated by color only, but in some cases it is difficult to distinguish each parameter. It would be nice to modify the picture to make it easier to check.

In revised manuscript we increased thickness of the lines corresponding to $G_F$. We hope it will simplify distinguishing of the lines.

2. The classification of aerosol types was explained in 4.4. In Fig. 14, dust, continental, and smoke were classified only by the relationship between particle depolarization ratio and fluorescence capacity. In addition to these two factors, it would be good to add a table that summarizes lidar ratio, EAE, and BAE and displays the average value for each type.

The main intensive parameters for days with high depolarization ratios are provided by Table 1. We didn't provide average values for aerosols in Fig.14, because in some way it may mislead: we observe mixture of pollen and background aerosol, thus average values will depend not only on pollen parameters but also on background aerosol contribution. We will try to indentify the pollen types and estimate their parameters during the next measurement campaign.

---

## Author Comment (AC2)

**Response to Referee#2**

First of all, we would like to thank Referee#2 for very careful reading the manuscript and for numerous useful suggestions.

**General comments:**
The authors present multiwavelength Mie–Raman–fluorescence lidar measurements during pollen season in France. The depolarization ratios at 3 wavelengths, and the fluorescence backscattering at 466 nm were used to characterize different aerosols (background, smoke, dust, and pollen mixture). Pollen are not yet as well characterized as other aerosol types, this study suggests a method of classifying pollen from other types using combined measurements of depolarization and fluorescence.
The dataset is interesting, and the manuscript is well written. However, I have two major concerns which should be addressed before publication.
**Specific comments:**
**My major concerns**:
1. The authors present a simulation in sect.3.1 (L156-174), to estimate the wavelength dependence on depolarization ratio of pollen. The results show a strong spectral dependence increasing with wavelength. They conclude that such increase of PDR with wavelength is an indication of the presence of pollen.
However, the presented simulation is based on too many assumptions. The conclusion here may mislead the reader about the wavelength dependence of pollen depolarization.
a) In fig. 2, the curve shape depends a lot on the assumptions you made. E.g., if I take the laboratory measurements in Cao et al. 2010 for birch pollen, depo355 = 0.08, depo532 = 0.33, depo1064 = 0.28, the simulated PDR1064 will be smaller than PDR532 when betaP(532)/beta(532)> 0.8. And if ÅE values are different from your assumptions, the curve slope/shape can change a lot.

[Figure]

Yes, reviewer is right, in this simulation numerous assumptions are made. Definitely, for $\dfrac{\beta^{p}_{532}}{\beta_{532}}$ close to 1.0, spectral dependence of depolarization ratio will be determined by the pollen

depolarizing properties. Hower for low contribution of pollen to backscattering (which is a typical situation), the depolarization ratio at 1064 will be the highest. In our manuscript we emphasize, that this can be specific feature, pointing to the presence of pollen. In the revised manuscript we added the passage:

The results demonstrate strong variation of spectral dependence for different taxa, and for most of the samples $\delta_{532}$ exceeded both $\delta_{355}$ and $\delta_{1064}$. In particular, for birch pollen, depolarization ratios at 355 nm, 532 nm and 1064 nm are of 8%, 33% and 28%. The Angstrom exponents of the background aerosol may also vary in a wide range, thus spectral dependence of depolarization ratio of aerosol - pollen mixture will differ from simplified modeling shown in Fig.2. However, for moderate contribution of pollen to the total backscattering ( $\frac{\beta_{532}^{p}}{\beta_{532}}$ below ≈0.5), the depolarization at 1064 nm will be higher than that at shorter wavelengths. Thus increase of the particle depolarization ratio with wavelength can be an indication of the presence of large, irregularly-shaped pollen grains. Using the depolarization ratios from Cao et al. (2010) in analysis, we should also keep in mind that measurements in the chamber were performed at low RH, and depolarization ratios at higher RH may be different.

b) L168, you only mentioned depo1064 exceeded depo355, but in Cao et al., for most pollen types, depo532 are higher than both depo1064 and 355.
It is corrected in revised manuscript. And we should remember, that their measurements were performed in the chamber at low RH, so in the real atmosphere the depolarizations can differ.

c) L172, "similar spectral dependence can be provided also by the dust particles". For coarse mode dust, laboratory studies show values: depo355=0.27, depo532=0.37, and depo1064=0.27. see review of Mamouri and Ansmann, 2017 and reference therein for more details.
Mamouri, R.-E. and Ansmann, A.: Potential of polarization/Raman lidar to separate fine dust, coarse dust, maritime, and anthropogenic aerosol profiles, Atmos. Meas. Tech., 10, 3403–3427, https://doi.org/10.5194/amt-10-3403-2017, 2017.

Yes, it is true, dust presents spectral dependence of depolarization, but for small dust contribution, depolarization at longest wavelength (1064 nm) should be the highest. In our measurements we observed this effect many times for ice crystals. Still, in the revised version of manuscript we removed this passage to avoid misunderstanding.

d) L308, "expected ratio depo1064/depo355 is about 2.3". I don't think you can use the simulated ratio as an expected value for the observations. In fig.8b-c, the observed ratio agrees with the simulation, but in fig.8e-f, fig.10 and fig.11, there is no such agreement.
We agree, it is too speculative. This passage is removed from revised manuscript.

Fig.10, depo532 almost equal to depo1064. Totally different than the simulation.
Simulation was performed for very model situation, to illustrate influence of background aerosol properties on spectral dependence of depolarization ratio in the presence of pollen. In fig.10 depolarization ratio at 1064 nm is below 15% ( $\frac{\beta_{532}^{p}}{\beta_{532}}$ below ≈0.5) and BAE is about 1.2 (we used 1.5 in simulation). Moreover, depolarization ratios of pure pollen at 532 may exceed

depolarization at 1064 nm. This is probably the reasons why $\delta_{532}$ and $\delta_{1064}$ in Fig.10 are rather close. We added to manuscript:

The depolarization ratios $\delta_{532}$ and $\delta_{1064}$ inside the PBL are close, which is in contrast with results in Fig.8, where $\delta_{1064}$ exceeds $\delta_{532}$. This difference can be due to the different types of pollen analyzed, which is probably grass in Fig.8 and birch in Fig.10. Besides, the BAE in Fig.10 is lower than in Fig.8, which decreases the influence of background aerosol on the spectral dependence of the depolarization ratio.

Fig.11, at 1250m, depol355=depo532 = 0.14. More discussions on those wavelength dependencies on depolarization ratio are needed.

In the episode shown in Fig.11, besides pollen the smoke particles presented. By 1250 m height contribution of smoke starts to increase, thus depolarization ratio at 355 nm rises (for smoke particles normally $\delta_{355}$ is the highest). We added to manuscript:

Depolarization ratio $\delta_{355}$ is about 15%, and this is higher than corresponding values shown in Fig.8, 10, which again may corroborate the presence of smoke particles. We should recall, that smoke particles are small, so, in contrast to pollen, their presence influences $\delta_{355}$ stronger than $\delta_{1064}$. The enhanced values of $\delta_{355}$ and $G_F$ were observed during 20-23 April period, indicating to the possible presence of the biomass burning particles in aerosol mixture.

e) L170-171, L282-283, L249-250, the related conclusions/descriptions should be checked.

We made corrections and added passage:

Such spectral dependence of depolarization ratio can be partly due to the contribution of the background aerosol, as follows from model calculation in Fig.2.

If you want to include such simulation in this study, more complete investigations should be done, and more discussions are needed here. I understand that your choice of parameters is not critical for presenting the approach. Nevertheless, it would be nice to get an idea of why those specific values have been selected (e.g. pollen depol, ÅE).

Fig.2 is modified and simulations for different BAE are added.

[Figure]

Fig.2. Depolarization ratios at 355, 532 and 1064 nm as a function of pollen contribution to the total backscattering coefficient $\dfrac{\beta^{p}_{532}}{\beta_{532}}$ . Depolarization ratios of pollen ($\delta^{p}$) and background aerosol ($\delta^{b}$) are assumed to be spectrally independent $\delta^{p}$=30% and $\delta^{b}$=3%. Backscattering coefficient of pollen is spectrally independent. The backscattering Angstrom of background aerosol exponents were assumed to be the same for both pairs of wavelengths ( $A^{\beta}_{355/532} = A^{\beta}_{532/1064}$ ), and results are shown for the values $A^{\beta}$=1.0, 1.5, 2.0.

2. Using depolarization ratio and fluorescence for the aerosol typing is of great value. The fluorescence capacity can be a very good indicator for pollen presence. Also, it is very interesting to show the wavelength dependency on depolarization ratios from observations, which could be one characteristic of pollen.
However, more information on pollen should be added: Are they really pollen? Which pollen type?
From fluorescence and polarization measurements we conclude that pollen present in aerosol mixture. Unfortunately at current stage we are not able to indentify the type of pollen.

a) L137, "pollen observed in the boundary layer by the lidar are probably transported from other regions." Short-range or long-range transported? The air mass origins are not always presented for some cases.
During the campaign, is there any good cases with local pollen? At the beginning of April, there were high pollen concentrations (mainly betula), together with hight fluorescence bsc. It will be very interesting if you have any good cases during that period. Characteristic values for different pollen types can vary a lot.
We could provide profiles of depolarization ratios and fluorescence starting from 500-750 m height and high pollen concentration at the ground didn't demonstrate strong correlation with fluorescence inside the PBL. This is why we think that pollen observed with lidar are transported from other regions. Basing on the back trajectories we are not capable to specify these regions at current stage. For analysis we choose days with low relative humidity, otherwise the effects of hygroscopic growth interfere and complicate the analysis. In particular, in the beginning of April we didn't have good cases. In our study we tried do demonstrate the potential of fluorescence – depolarization measurements, but definitely this is only the first step.

b) L126-139, you discussed about the links with high fluorescence with presence of pollen at ground level, but as the conclusion you said no direct correlation. It is a bit confusing.
As follows from Fig.1, there is some correlation between ground and lidar measurements. In particular, we didn't see strong fluorescence before the pollen season. Still, the highest fluorescence was in the end of March, while in the beginning of April, when birch pollen was of high concentration, the fluorescence was lower. We can not exclude also, that different pollen may have different fluorescence capacity.

c) You have chosen observations with high fluorescence and high depolarization for pollen study, what if there is smoke and dust mixture?

Backtrajectories didn't demonstrate the dust transport for cases presented. Theoretically, may be it is possible to construct dust/smoke mixture, which will mimic the optical properties of pollen. However this question demands a separate consideration.

**My main comments:**
3. The full overlap is ~1000 m (sect.2.1). Have you performed the overlap correction on lidar signal? if yes, what was the lower limit after the correction?
No, we didn't perform overlap correction. To decrease the start height, some observation sessions were performed at an angle of 30 deg to horizon. Corresponding sentence is added to the manuscript.

a) L130, fluorescence bsc, maximal values in 500-1000 m range, do you mean "range" or "height" here? These values are after the overlap correction?
This is height. Correction was introduced in manuscript.

b) Fig.6, optical profiles at 500-1000 are reliable? with an overlap correction?
Backscattering coefficient and fluorescence backscattering can be calculated down to 500 m. Besides, the range 500-1000 m was not essential for the case considered, because we focus on the dust layer above 2000 m.

4. L145, "the pollen loading in the north of France is significantly lower", this is not correct. Regarding fig.1, daily concentration of birch pollen is similar as 2h concentration reported in (bohlmann et al., 2019), ~3000 m-3 for the peak value; and much higher than pollen counts reported in Sassen et al. 2008. If you take 2h pollen concentration, you may have even higher values.
I suggest you plot the 2h pollen concentration instead of daily one, which provide more information. High pollen concentrations are often observed during daytime, but it can also present at night.
Yes, pollen concentration at ground level was quite high, but we never observed the clouds of birch pollen with depolarization of 30% as Sassen did. The maximal depolarization at 532 nm was slightly above 15%, this is why we conclude that pollen concentration at 1000 m height was lower than over Alaska of Finland.

**Technical corrections:**
- L40, 694 nm, not 0.694 nm.
  Corrected
- L44, please also give the region of the mentioned pollen studies.
  Corrected
- L50, a recent paper of bolmann et al. 2021 on depol ratio at 3 wavelengths is worth to be mentioned in the introduction.
  Bohlmann, S., Shang, X., Vakkari, V., Giannakaki, E., Leskinen, A., Lehtinen, K., Pätsi, S., and Komppula, M.: Lidar Depolarization Ratio of Atmospheric Pollen at Multiple Wavelengths, Atmos. Chem. Phys. Discuss. [preprint], https://doi.org/10.5194/acp-2020-1281, in review, 2020.
  Added
- L144 change 2011 to 2019.

Corrected
- L197, fig.1 not 1a
  Corrected
- L239, "suggesting that this layer may contain smoke or pollen particles". From the figure, depo1064 of the layer at 1600 m is about 0.08, depo1064 of the previous smoke case is only 0.015, Can you state on this? Back trajectories for layers of this case (sect.3.4)?
  Yes, basing on relatively high depolarization ratio at 1064 nm we assume that this is probably pollen. Still the layer is very weak and the rest of intensive parameters are not available, so we refrain from ultimate statement. Because, in principle, as reviewer has already mentioned, it can be mixture of smoke and dust. Some organic particles should present in this layer, because fluorescence capacity is high. This is why we write "pollen or smoke'. And the layer is too weak to be identified from BT.
- L263, was there a RH threshold you used?
  We tried to skip the sessions with RH above approximately 60% in the height range of interest.
- L285, pollen are medium to high absorption particles in the literature, how to explain the difference on low lidar ratio with literature?
  Lidar ratios in our measurements varied in a wide range. In some cases lidar ratio was quite low. May be absorption of pollen for that cases was low or it can be the effect of water uptake. We don't have ultimate explanation.
- L288, EAE=2, fine mode particles are dominant.
  Yes, EAE is between 1.75 and 2.0, so fine particles are dominant and contribution of pollen is not high. For such depolarization similar EAE was reported also by Bohlmann et al. (2019).
- L326, volume depolarization at 1064 is above 15%? Please check. In fig.10, particle depol. 1064<0.15
  In Fig.9 volume depolarization is above 15% for some short periods, but Fig.10 shows averaged values, so it is below 15%. In the manuscript we changed "above 15%" for "about 15%".
- L345, what is the typical value for the pollen?
  As follows from Fig.14, fluorescence of aerosol mixture depends on pollen contribution. For typical mixture with depolarization at 532 nm in 10%-15% range, the capacity was $(1.0 – 1.6)*10^{-4}$.
- L356, unit for beta is "Mm-1 sr-1"
  Yes, corrected
- L357, VDR at 1064 nm? For the layer, VDR1064>0.1 but PDR1064 is only 0.08 (L362)?
  In Fig.13 we show averaged values, but peak values of volume depolarization exceeded 12%, we write about it lower. In manuscript we again changed "above 10%" for "about 10%".
- L383, how do you define: "when the contribution of pollen to the total particle backscattering was significant."
  We took days with depolarization at 532 nm above 10%, which is higher than depolarization of background aerosol.
- L399, for smoke, "high Gf" not "low Gf"
  Yes, it was mistake. Corrected.
- L408, if you assume pure pollen depolarization ratio of 0.2-0.5, how Gf values range?

We provide fluorescence capacity only for aerosol mixture. We assumed that pollen depolarization at 532 is about 30%, so pure pollen $G_F$ should be about $2.5*10^{-4}$. But if depolarization is 50% then $G_F$ should be about $4.0*10^{-4}$. But the only way to verify it is to perform measurements in the region, where pollen contribution is predominant.

- Fig.1, you can add the presented cases in the figure.
  Added
- Fig.6, in x-label and the caption, beta1064 or beta532?
  Done
- Fig.7, add RH profile for the hygroscopic growth study.
  Measurements of RH by radiosonde were not collocated with lidar observations, so we can make only qualitative conclusion about RH increase. This is the reason why we wouldn't like to provide RH profile.
- Fig.10, please provide in the caption the depol values you used for the retrieval of the pollen bsc coefficient.
  Done

---

## Author Comment (AC3)

First of all we would like to thank the Reviewer for careful reading the manuscript and for useful comments.

The paper contains new and complex lidar observations of pollen. A triple-wavelength polarization Raman lidar with aerosol fluorescene channel is built up and mixtures of pollen and anthropogenic pollutionare measured, as well wildfire smoke and dust outbreak events.
The paper is well written and clearly worthwhile to be published.
I have only minor comments.

P2, L45-49: The Finish group only measured lidar ratios? No depolarization ratios?
They presented recently in ACPD the pollen polarization measurements at 3 wavelengths. Corresponding reference and comment are added to the manuscript

P3, L63: LILAS is the abbreviation for…? Lille lidar for atmospheric studies …?
Added

P4, L95: How do you overcome the problem with the 1064 nm backscatter reference value? Are you using cirrus backscatter and set beta 532 = beta 1064? Otherwise, beta 1064 is rather uncertain! Please comment on that!
Yes, calculation of backscattering at 1064 is challenging. For the cases presented in the manuscript we were able to choose height range in the high troposphere, which was free of clouds and aerosols. Thus, molecule scattering was used as reference. We assumed that aerosol contribution is insignificant when lidar signal matched profile of molecular scattering and (simultaneously) the volume depolarization at 1064 nm was below 1%.

P4, L106: It is a pity that you had to remove the water vapor channel, and at the same time, RH is an important parameter in your study… , and you have to make use of radiosonde observations far away. In case of good observations of water vapor mixing ratio profiles, one can easily and accurately derive RH profiles by using weather model temperatures in addition.
Yes, water vapor is very important. Analyzing the data we used radiosonde data and the model also. Unfortunately only qualitative analysis of RH was possible. At present, the water vapor channel in the lidar is recovered.

P5, L125-138: I would prefer a table (maybe even in Figure1) with all the specific names for the substances (betula, …, poaceae…) and the translations in addition Quercus (oak) , Poaceae (grass), betula (birch) and so on, if that is possible…
We added translations of pollen to Fig.1.

P7, L194: I find that depolarization ratios of 5-7% are quite high! What is the reason, is that specific for Lille? Is that the remaining pollen impact.
Yes, we think this is contribution of remaining pollen. In June amount of pollen decreases but it still exists.

P8, L215-217: But usually smoke layers show low depol ratios of $<$0.05 at all wavelengths in the lower troposphere as the example in Haarig et al., Canadian smoke paper in ACP, 2018, shows. An exception is the observation shown in Burton et al. 2015, not the rule.

Yes, such strong dependence is more typical for high troposphere. However during September 2020 we observed numerous cases with strong spectral dependence of smoke depolarization in low troposphere. Corresponding publication is in preparation. We have added passage to the manuscript:

We should recall also, that increase of the particle depolarization ratio at 355 nm is more typical for the aged smoke layers in the high troposphere (Haarig et al., 2018), though we observed this increase at lower altitudes over Lille during smoke episodes in Summer – Autumn 2020.

P10, L289: BAE strongly depends on particle refractive index and shape, and EAE? only weak dependence , or even no dependence?... what do you mean here… ?

EAE has very weak dependence on the refractive index, so it is sensitive to the size only. It was discussed for the case of dust in our recent paper of Veselovskii et al., (Atm. Chem. Phys., 20, 6563-6581, 2020). EAE is also not sensitive for particle shape: computations for spheres and spheroids bring to the same result.

P14, L398: smoke – low depol … high GF

Corrected

P26, Fig 5: more than four hours of signal averaging! How sensitive are the results to changes in the aerosol conditions?

Smoke layer is very stable and results are not sensitive the choice of averaging interval. So we prefer to show averaged over night profiles.

P27, Fig 6: x-axis text starts with beta-1064, but shown is the beta-532 backscatter (in green)

Corrected

P29, Fig 8, again 5 hours of signal averaging! Please comment on signal averaging, and that you need stable conditions.

Corresponding comment is added.

The atmospheric conditions for these nights were stable so the profiles presented are averaged over approximately five hours.

P31, in Fig.9 very variable aerosol structures are visible, but in Fig.10 all nine hours are averaged. Please provide a comment on the impact of aerosol variability on the retrieval products.

Yes, 9 hours of averaging is a lot. Actually this interval could be decreased. But inside the PBL the variations of lidar signal and the fluorescence backscattering were actually not so strong, so we think that averaged over night profiles of particle parameters are representative. Corresponding comment is added to the manuscript.